# Biosynthesis and Characterization of ZnO Nanoparticles Using *Ochradenus arabicus* and Their Effect on Growth and Antioxidant Systems of *Maerua oblongifolia*

**DOI:** 10.3390/plants10091808

**Published:** 2021-08-30

**Authors:** Hassan O. Shaikhaldein, Fahad Al-Qurainy, Salim Khan, Mohammad Nadeem, Mohamed Tarroum, Abdalrhaman M. Salih, Abdel-Rhman Zakaria Gaafar, Aref Alshameri, Saleh Alansi, Norah Arrak Alenezi, Norah S. Alfarraj

**Affiliations:** Department of Botany, College of Science, King Saud University, P.O. Box 2455, Riyadh 11451, Saudi Arabia; fal_qurainy@hotmail.com (F.A.-Q.); salimkhan17@yahoo.co.in (S.K.); mohammadnadeem911@hotmail.com (M.N.); med_taroum@yahoo.fr (M.T.); abdalrahamanm@gmail.com (A.M.S.); abdou.gaafar@yahoo.com (A.-R.Z.G.); arefshamiry@yahoo.com (A.A.); alansi1975@yahoo.com (S.A.); noalenzi@iau.edu.sa (N.A.A.); 438203416@student.ksu.edu.sa (N.S.A.)

**Keywords:** ZnO NPs, green-synthesize, *M. oblongifolia*, chlorophyll, proline, antioxidant enzymes

## Abstract

Zincoxide nanoparticles (ZnO NPs) are among the most produced and used nanomaterials worldwide, and in recent times these nanoparticles have also been incorporate in plant science and agricultural research. The present study was planned to synthesize ZnO NPs biologically using *Ochradenus arabicus* leaves and examine their effect on the morphology and physiology properties of *Maerua oblongifolia* cultured in vitro. ZnO NPs were characterized by UV–visible spectroscopy (UV–vis), X-ray diffractometer (XRD), Fourier transform infrared spectroscopy (FT-IR), and transmission electron microscopy, which demonstrated hexagonal shape nanoparticles of size ranging from 10 to 50 nm. Thus, the study uncovered an efficient, eco-friendly and simple technique for biosynthesis of multifunctional ZnO NPs using *Ochradenus arabicus* following growth of *Maerua oblongifolia* shoots in different concentrations of ZnO NPs (0, 1.25, 2.5, 5, 10, or 20 mg L^−1^) in Murashige and Skoog medium. Remarkable increases in plant biomass, photosynthetic pigments, and total protein were recorded up to a concentration of 5 mg L^−1^; at the same time, the results demonstrated a significant reduction in lipid peroxidation levels with respect to control. Interestingly, the levels of proline and the antioxidant enzyme catalase (CAT), superoxide dismutase (SOD), and glutathione reductase (GR) activities were increased significantly in response to all ZnO NP treatments. These findings indicate that bioengineered ZnO NPs play a major role in accumulation of biomass and stimulating the activities of antioxidant enzymes in plant tissues. Thus, green-synthesized ZnO NPs might be of agricultural and medicinal benefit owing to their impacts on plants in vitro.

## 1. Introduction

Nanotechnology nowadays is the focus of scientific community interest and has taken hold in all fields of science due to the necessity of applications of nanomaterials in many aspects of human endeavor, such as industry, business, medicine, public health, and agriculture [1]. In general, nanotechnology comprises synthesis of nano-sized (1–100 nm) particles. Particles of this size are referred to as engineered nanomaterials (ENMs) [2]. While many researchers have stated various techniques for the manufacturing of metal oxide nanoparticles (NPs), biological synthesis using plant extracts and microorganisms is simpler, less costly, and more eco-friendly as compared to physicochemical procedures [3]. Additionally, the NMs synthesized using plants are more steady, less toxic, and biocompatible [4]. Many studies have been carried out on the vastly used nanomaterials (NMs), such as fullerenes, ZnO, TiO_2_, CuO, and Ag [5]. Among these important nanomaterials, zinc oxide (ZnO) engineered nanoparticles (ENPs) have a unique position [6]. In manufacturing industries worldwide, ZnO NPs are among the most produced and used, in recent times these nanoparticles have also been incorporated in plant science and agricultural research, although contradictory results on their benefits are reported [7]. A variety of synthetic methods are used for the synthesis of ZnO NPs; these methods can be broadly divided into three types; that is, chemical, physical, and biological techniques [8]. ZnO NPs have been effectively synthesized using the biological technique. The main idea behind the green synthesis of ZnO NPs is that the natural materials (plants and microorganism) contain phytochemicals which act as both reducing as well as stabilizing (capping) agents. They reduce the metal (zinc) to the 0-valence state and then through calcinations, oxide may be added to the metal [9]. Different examples have been reported for synthesis of ZnO NPs using bio-extracts and their applications in plant species such as fodder maize (*Zea mays*) [10], *Allium cepa* [11], and *Sesamum indicum* [12]. At certain concentrations in plant cell cultures, ZnO NPs are reported to play an essential role in enhancing growth, seed germination, photosynthetic efficiency, chlorophyll content, starch content, and notably secondary metabolites production [13,14,15,16,17].

*Maerua oblongifolia* is a rare plant found in Saudi Arabia. The plant is a member of the family Capparaceae. It used to cure diseases such as stomach ache, fever, cough, and skin infections, urinary calculi, diabetes, epilepsy and abdominal colic [18]. Because of overexploitation for feed, food, lumber, and medicinal usages as well as its slow regeneration rate, wild populations of this plant are quickly decreasing [19]. Therefore, there is a considerable demand to enhance the regeneration of *M. oblongifolia* with micropropagation [20]. This can be attained successfully with the application of NPs [19].

Reports pertaining to the morphophysiological characteristics of plants under exposure of synthesized ZnO NPs are scarce [21]. Hence, in the present investigation it was planned to synthesize and study the influence of ZnO NPs on the regeneration, biomass, and antioxidant enzyme activities of *M. oblongifolia* raised in vitro.

## 2. Material and Methods

### 2.1. Synthesis of ZnO NPs

The ZnO NPs were synthesized using a method reported by Al-Shabib et al., (2018) [22] with some modifications. The tissue culture lab of the King Saud University, Riyadh, Saudi Arabia, provided *Ochradenus arabicus*. Five grams of fresh leaves were thoroughly cleaned with running tap water, subsequently with deionized Milli-Q water, and then finely chopped, and soaked in a round-bottom flask containing 100 mL of deionized water. The mixture was boiled at 60 °C for 10 min. The leaf extract was allowed to cool to room temperature, filtered in a 100 mL Erlenmeyer flask through Whatman number-1 filter paper, and the filtered was preserved for further experimental use.

The chemicals, zinc acetate (ZnC_4_H_6_O_4_) and sodium hydroxide (NaOH), were purchased from Sigma-Aldrich Chemical Corp. Two millimolar zinc acetate and 0.2 mM of NaOH were dissolved in 200 mL Milli-Q water. Then, 100 mL of leaf extract, zinc acetate and NaOH mixture were mixed together and left in a stirrer (Janke & Kunkel, IKA-Labortechnik, Staufen, Germany) for 24 h. The color of the reaction mixture changed to faint yellowish, indicating the initial synthesis of ZnO NPs. Then, precipitate was separated from the reaction solution using Whatman number-1 filter paper and dried using a hot air oven (Vaciotem-T oven, JP Selecta SA, Barcelona, Spain) operating at 48 °C for one day. Calcination of the powder was performed at 600 °C via furnace (DKN 602, Yamato Scientific Co., Ltd., Tokyo, Japan) for 3 h to obtain ZnO NPs. Finally, the product was collected and kept in airtight bottles for further studies.

### 2.2. Characterization of ZnO NPs

Optical properties of biosynthesized ZnO NPs were characterized using different techniques of analysis. First, ultraviolet–visible (UV–Vis) spectroscopy (Shimadzu, Tokyo, Japan) was monitored to check the reduction technique used for ZnO NP synthesis. Fourier-transform infrared spectroscopy (Termo Scientific Nicolet 6700 FT-IR spectrometer) was used to detect the presence of potential biomolecules and functional groups. X-ray diffraction (Rigaku Ultima IV, Germany XRD) was used to check the formation, crystalline behavior, and quality of the bioreduced ZnO NP powder. Crystalline size of the synthesized ZnO NPs was estimated using Scherrer’s formula. The shape and size of the synthesized ZnO NPs was confirmed using transmission electron microscopy (TEM) (JEM-1011; JEOL Ltd., Tokyo, Japan).

### 2.3. Plant Materials

*Maerua oblongifolia* specimens were collected from the southern parts of Saudi Arabia and germinated in vitro via micropropagation in Murashige and Skoog (MS) media using the method developed by Al-Qurainy, et al. [23]. In the tissue culture laboratory of the King Saud University.

The experiment was achieved in the tissue culture laboratory of the King Saud University. Various concentrations of ZnO NPs (0 mg L^−1^, 1.25 mg L^−1^, 2.5 mg L^−1^, 5 mg L^−1^, 10 mg L^−1^, and 20 mg L^−1^) were added to the MS media and three-centimeter-long *M. oblongifolia* shoots were transplanted into Magenta boxes (GA-7). Five explants were placed in each box containing 100 mL MS media. The tests for each treatment were performed three replications.

Samples were harvested after 45 days of culture to analyze the plant performance in terms of the following growth parameters: shoots number per explant, shoot length per explant, leaves number per explant, fresh weight, and dry weight.

### 2.4. Estimation of Photosynthetic Pigment Content

For determination of the photosynthetic pigments chlorophyll a (chl a), chlorophyll b (chl b) and carotenoid were estimated as per the method described by Arnon [24]. Leaves (0.1 g) were homogenized in chilled 80% acetone. The samples were kept at −4 °C for 24 h before the blend was transferred to a 2 mL Eppendorf tube. Finally, absorbance of supernatant was tested using a UV-1800 spectrophotometer (Shimadzu, Japan) at 645 and 663 nm for chlorophyll content and at 470 nm for carotenoids.

### 2.5. Estimation of the Total Soluble Protein Content

The total soluble protein content was evaluated as per the methodology reported by Bradford [25]. We homogenized 300 mg in a 1 mL phosphate buffer. Equal volumes of supernatant and TCA were mixed and centrifuged; the pellet was dissolved in 1 mL of 0.1 N NaOH. The absorbance was estimated photometrically at 595 nm with bovine serum albumin as the standard. The protein content was expressed as mg/g of the fresh weight.

### 2.6. Estimation of the Proline Content

The proline content was estimated as per the method described by Bates et al. [26]. Fresh leaves (0.4 g) were homogenized in 10 mL of 3% aqueous sulfosalicylic acid. Subsequently, the blend was centrifuged; 2 mL of supernatant was transferred into a test tube, and 2 mL of ninhydrin and 2 mL of glacial acetic acid were added. Thereafter, the mixture was heated up to 100 °C for 1 h. Following boiling, the reaction was terminated by placing the tubes for 5 min in an ice bath. Then, we added 6 mL of toluene was to each tube and mixed strenuously for 15 s. The absorbance of the upper phase was measured at 520 nm using a UV-1800 spectrophotometer (Shimadzu, Kyoto, Japan). The proline content was expressed as μg/g fresh weight.

### 2.7. Lipid Peroxidation Content

Malondialdehyde (MDA) was analyzed by measuring the production of thiobarbituric acid reactive substances (TBARS) using TBARS assay. The MDA content in leaves was determined according to method described by De Vos et al. [27]. The plant tissues were homogenized in 5 mL of 0.1% trichloroacetic acid (TCA) and the material was subjected to centrifugation using an Eppendorf 5417R centrifuge at 10,000 rpm for 5 min. Then, 1 mL of clear supernatant was put in a separate test tube, to which 4 mL of 0.5% thiobarbituric acid (TBA, in 20% TCA) was added, and the mixture was incubated at 95 °C in a water bath for 30 min. It was quickly cooled in an ice bath and centrifuged at 5000 rpm for 5 min; the absorbance of the supernatant (containing MDA) was read at 532 nm and corrected to unspecific turbidity by subtracting the value at 600 nm on a UV–Vis spectrophotometer. The blank was 0.5% TBA in 20% (*w*/*v*) TCA.

### 2.8. Enzyme Extraction and Estimation of the Enzyme Activity

Enzyme extraction and determination were employed using the methodology reported by Jogeswar et al. [28]. Fresh leaf tissues of *M. oblongifolia* were initially crushed in liquid nitrogen and dissolved in 100 mM sodium phosphate buffer (pH 7.4) that contained 0.1 mM ethylenediaminetetraacetic acid, 1% (*w*/*v*) polyvinylpyrrolidone, and 0.5% (*v*/*v*) Triton-X 100. The homogenous mixture was centrifuged at 10,000 rpm for 10 min at 48 °C to obtain the supernatant.

Superoxide dismutase (SOD, EC 1.15.1.1) activity was assayed as per the method reported by Marklund and Marklund [29]. The reaction mixture contained 1 mL of 0.25 mM pyrogallol, 1.9 mL of 0.1 M sodium phosphate buffer (pH 7.4), and 100 μL of enzyme extract. The absorbance was recorded at 420 nm. The SOD activity (U/g protein) was defined as the amount of enzyme needed for 50% inhibition of pyrogallol oxidation.

The catalase (CAT, EC 1.11.1.6) activity was read by measuring the absorbance at 240 nm, according to the method developed by Claiborne [30]. The reaction mixture contained 1 mL of 0.059 M H_2_O_2_ in 0.1 M sodium phosphate buffer (pH 7.4), 1.9 mL of distilled water, and 100 μL of enzyme extract. The CAT activity was expressed as unit/g of protein.

Glutathione reductase (GR) (EC 1.6.4.2) activity was read by the procedure described by Schaedle and Bassham [31]. The reaction mixture was following oxidation of nicotinamide adenine dinucleotide phosphate (NADPH) at 340 nm (e, 6.2 mM_1 cm_1) for 180 s in 2 mL of an assay mixture containing 50 mM potassium-phosphate buffer (pH 7.2), 3 mM Na2EDTA, 0.15 mM NADPH, 0.5 mM GSSG, and 100 mL of enzyme extract. GR activity was expressed as EU mg^−1^ protein.

### 2.9. Statistical Analyses

A completely randomized experimental design was used. Statistical analyses were employed using one-way analysis of variance and comparison was performed using Duncan’s new multiple range test (*p* ≤ 0.05) in SPSS v. 20 for Windows.

## 3. Results

### 3.1. Green Synthesis and Characterization of ZnO NPs

The reducing of *O. arabicus* leaves extract caused a visible color change on stirring for 24 h; the color change of the solution from white to pale yellow was the preliminary evidence of synthesis of ZnO NPs.

### 3.2. UV-Vis Spectroscopy

UV-Vis spectroscopy is an ideal method that is usually performed for the confirmation and characterization of the synthesis of ZnO NPs based on surface plasmon resonance (SPR) [32]. The optical absorption spectrum of the synthesized ZnO NPs was determined in the range of 200–800. UV-Vis spectroscopy analysis of the synthesized ZnO nanoparticle suspension illustrated a characteristic absorption peak at 380 nm as shown in Figure 1a.

### 3.3. XRD Analysis

XRD spectra delivers an insight about the crystallinity of nanoparticles [33]. Figure 1b shows XRD spectra of the synthesized ZnO NPs using *O. arabicus* leaf extract. Synthesized ZnO NPs presented diffraction peaks at 2θ values 31.81°, 34.45°, 36.27°, 47.54°, 56.68°, 62.87°, 66.32°, and 67.92° corresponding to the lattice plane of (100), (002), (101), (102), (110), (103), (112), (201), and (220), respectively. The appearance of peaks was appropriate to ZnO NPs structure. The crystalline size of the nanoparticles was calculated using Scherrer’s equation and was found to be 35 nm.

### 3.4. FTIR Spectroscopy

FTIR spectrum analysis was performed to determine the surface chemistry of the ZNO NPs and applied to obtain the details of functional groups involved in the biomolecules responsible for reducing and capping the bio-reduced ZnO NPs [34]. The FTIR spectrum results relatively revealed a complex of bioorganic compounds in the plant extract, which are responsible for the reduction and stabilization of ZnO NPs. The FT-IR profile exhibited four peak positions at 704, 1634, 2078, and 3400 cm^−1^ as demonstrated in Figure 1c.

### 3.5. Electron Microscopy

The morphological analysis of the synthesized ZnO NPs was completed using TEM. The TEM image revealed that the synthesized ZnO NPs were hexagonal in shape and had various sizes ranging from 10 to 50, as given in Figure 1d.

### 3.6. Effects of ZNO NPs on In Vitro Shoot Reproduction

The growth of *Maerua oblongifolia* at the different concentrations of ZnO NPs was studied according to the morphological traits such as fresh weight, dry weight, shoot number, shoot length, and leaf number (Figure 2). A dramatic increase in all traits was observed due to ZnO NP exposure compared to control (Table 1). Plants exposed to 5 mg L^−1^ ZnO NPs promoted all morphological traits. However, the control group demonstrated the least development in all morphological traits.

### 3.7. Effects of ZnO NPs on Photosynthetic Pigments, and Total Soluble Protein

Figure 3 shows the photosynthetic pigment content in leaves of *M. oblongifolia* plants supplemented with different doses of ZnO NPs. Results show that the levels of photosynthetic pigments (chl. a, b, and carotenoids) were significantly increased by ZnO NP treatments compared to the control group. A ZnO NP concentration of 5 mg L^−1^ yielded the highest levels of chl. a and chl. b, while the control group had the lowest chl. a and chl. b levels. The highest level of carotenoids was recorded at both 2.5 and 5 mg L^−1^.

The level of total soluble protein contents in leaves of *M. oblongifolia* plants differed significantly as per the ZnO NP treatments to which the plants were exposed (Figure 3d). It is notable that the highest level of total soluble protein content resulted after supplementation with 5 mg L^−1^, while control treatment registered the lowest total protein level.

### 3.8. Effect of ZnO NPs on Proline Content

Proline content of the plants differed significantly as per the ZnO NP concentrations to which the *M. oblongifolia* plants were treated. Low concentrations of ZnO NPs (1.25, 2.5, and 5 mg L^−1^) showed a stable proline level while high concentrations of ZnO NPs (10 and 20 mg L^−1^) resulted in high proline levels. The highest proline level was in plants exposed to 5 mg L^−1^ of ZnO NPs (Figure 4a).

### 3.9. Lipid Peroxidation

The results (in Figure 4b) show the effects of ZnO NPs on the concentration of MDA as lipid peroxidation substances in the shoots of *M. oblongifolia*. Our results indicated that ZnO NP treatment decreased the level of MDA content over the control in the plants, and the control group recorded the highest level of MDA, while the lowest values were registered in the highest concentration where 20 mg/L of ZnO NPs were applied.

### 3.10. Effects of ZnO NPs on Antioxidative Defense Enzyme Activity

We analyzed the SOD, CAT, and GR activities to evaluate the effect of ZnO NPs on the activities of enzymes that were related to oxidative stress. The activity of the three enzymes were stimulated in plants as per increasing the concentrations of ZnO NPs, where 20 mg L^−1^ recorded the highest level of antioxidant enzymes (Figure 5a–c).

## 4. Discussion

Characterization of synthesized ZnO NPs is commonly performed using UV–Visible spectroscopy, XRD, FTIR spectroscopy, and TEM microscopy. These techniques provide the information on the formation, size, structure, and elemental composition of nanoparticles. Optical properties of nanosized particles is being commonly assessed by UV–visible absorption spectroscopy [35]. The absorption peak (380 nm) found in the present study obviously demonstrates the presence of ZnO NPs in the reaction mixture and it is in agreement with the earlier results of [33,36]. The FTIR results demonstrated a band around 3400 cm^−1^, potentially resulting from OH stretching vibrations; meanwhile, the peak at 2078 cm^−1^ for C=H suggests a strong stretch assigned to the alkyl methylene group. The peak at 1634 cm^−1^, corresponding to amide I, appears to be caused by carbonyl stretching in proteins [37]. The peak at 704 cm^−1^ was for the C=H bond assigned to strong a mono-substituted aromatic benzene group [38]. Bearing in mind the FWHM of the plane in (101), the crystalline size of the engineered ZnO NPs was recorded using Scherrer’s formula; and the average particle size of the sample was found to be 351 nm. XRD pattern analysis proved the characteristic hexagonal wurtzite crystalline structure of ZnO NPs, which is in line with the earlier result reported by [39]. TEM was used to characterize the shape and size of ZnO NPs that were synthesized using *O. arabicus* leaf extract. The TEM images clearly showed that the synthesized ZnO NPs were almost hexagonal in shape, with the average diameter of nanoparticles ranging from 10–50 nm approximately, in accordance with the earlier result reported by [35]. In general, XRD is mostly used to estimate the particle size of nanoparticles, however TEM is the preferable technique for the measurement of nanoparticle size. The Scherrer formula for calculating particle size gives an average value of the entire particle responsible for diffraction,.while when using TEM, besides directly determining particle size, the morphology of the particles can also be noted [40].

To determine the growth enhancing effects of ZnO NPs applied to the culture medium of *Maerua oblongifolia* plant, morphological characteristics such as fresh weigh, dry weight, shoot length, shoot number, and leaf number were studied. Exposure of ZnO NPs to in vitro shoots of the *M*. *oblongifolia* significantly boosted the vegetative growth in the plant, including improving plant height, number of plants per pot, and plant biomass. This enhancement can most likely be attributed to the role of Zn in the production of tryptophan—the precursor of indole-3-acetic acid phytohormone [41]. In addition, ZnO NPs can alter the phytohormone biosynthesis of cytokinins and gibberellins, which can drive to an expansion in the number of internodes per plant [10]. Concentration at 5 mg L^−1^ ZnO NPs achieved the greatest growth of all morphological attributes, while the growth was reduced at 10 and 20 mg L^−1^. This decrease in growth it might be due to a higher concentration of nanoparticles reaching toxic levels in stem and leaves, which reduced the plant growth; in an earlier study high concentrations of ZnO NPs drastically affected the growth of tomato plants [7]. The positive effect of ZnO NPs exposed to plants was also reported in wheat [42], cotton [38], cluster bean [43], and ryegrass (*Lolium perenne*) seedlings at 2 mg L^−1^ [44]. Silver nanoparticles and other nanomaterials’ exposure have also been reported to enhance the growth of *M. oblongifolia* [19]. Iron-based NPs improved the growth of maize [45]. Contrary to our findings, ZnO NPs were reported as being toxic in several plants species such as *Allium cepa* roots [46], rice [47], and wheat seedlings [48]. These contradictions might be closely related to the chemical composition, chemical structure, particle size, and surface area of the NPs [49,50]. The response of the photosynthetic pigments’ content (chl. a, chl. b and carotenoids) to ZnO NP treatment in the present study correlates with earlier reported findings [51] in which chlorophyll and other photosynthetic pigments in cilantro (*Coriandrum sativum*) were remarkably increased after application of ZnO NPs. Similar results were found in several plant species, according to [16] and [52], and the exposure of the ZnO NPs improved the photosynthetic pigments and protein in *Phaseolus vulgaris* and *Lupinus termis*. The reason behind the enhanced chlorophyll and other photosynthetic content in our study is most likely due to the presence of zinc as a vital nutrient for the plants, essential for protochlorophyllide formation. Zn plays an essential role in plant metabolism by influencing the activities of key important enzymes, such as carbonic anhydrase [51]. In addition, metal nanoparticles are powerful amplifiers of photosynthetic effectiveness that in parallel cause light absorption by chlorophyll, as they cause the transfer of energy from chlorophyll to nanoparticles [53,54]. Carotenoids act as antioxidant compounds soluble in plant cells. These compounds through a non-enzymatic pathway function to reduce oxidative damage to the plant. Carotenoids represent a vital class of antioxidant molecules, which are known to scavenge harmful free radicals, as well as protecting light-harvesting complex proteins and thylakoid membrane stability. In this study, it seems a certain amount of zinc induced oxidative stress increased carotenoid content synthesis [55]. Conversely to our findings, there are some studies which reported that ZnO NPs decrease the chlorophyll content in some plants species, such as kidney bean (*Phaseolus vulgaris*) and soybean (*Glycine max*) [56,57]. This ambiguity may be because some plant species may have different responses to Zn exposure than others.

Application of ZnO NPs at the concentrations in our study caused an increase in total protein content compared to control: 5 mg L^−1^ concentration recorded the highest increase in protein, which may suggest the initiation of de novo synthesis of the enzymes [58]. On the other hand, protein level started decreasing at doses of 10 and 20 mg L^−1^ respectively. The increase in protein at certain concentrations indicates the optimum dose limit for the growth of *M. oblongifolia* plants. However, the reduction in protein beyond this concentration indicates the toxic effect of ZnO NPs. Similar results have been reported by a study on lettuce [59].

Lipids are a key components of cell membranes. They are sensitive to oxidation processes, and generating lipid peroxides is an indicator of an increase in production of toxic oxygen species [60]. MDA content is considered an indicator of oxidative damage. Our findings showed that MDA was diminished in ZnO NP treated plants. It is known that Zn has the ability to stabilize and protect the biomembranes against peroxidative and oxidative stress, integrity of plasma membrane loss, and change in the permeability of plasma membrane [61]. Our results at in line with the findings of previous reports in *Vicia faba* [62], soybean [17], and wheat [42].

The process of oxidative damage is due to the imbalance process of reactive oxygen species (ROS) metabolism in plants. Antioxidant enzymes, such as SOD, CAT, and GR, are known to be the major protective factors protecting against ROS in plants, through which plants can scavenge H_2_O_2_ (hydrogen peroxide) and O_2_^−^ (superoxide radical) and other ROSs [27,63,64]. The activity of the antioxidant enzymes is significantly enhanced in plants upon exposure to ZnO NPs in a concentration dependent manner. In this study, SOD, CAT, and GR activities of *M. oblongifolia* plants exposed to ZnO NPS significantly increased with increasing of the doses. Boosted activity of antioxidant enzymes by supplementation of ZnO NPs may scavenge H_2_O_2_ and mitigate mineral uptake, reducing plant oxidative stress [65]. In addition, increasing CAT, SOD, and GR activity, notably with high concentrations of ZnO NPs, probably signals that the antioxidant enzyme system is adapting to counteract excessive production of ROS [66]. Our findings are consistent with the earlier studies documenting that the SOD, CAT, and GR activities were increased with increasing concentrations of ZnO NPs in rice seed [67]. Similarly, a previous trial documented that CAT, SOD, and GR activities were enhanced in *Arabidopsis thaliana* seedlings treated with gold nanoparticles [1]. Hence, enhanced antioxidant potential of *M. oblongifolia* plants under ZNO NPs treatment signifies better performance. Lu et al. [68] also observed that SOD, CAT, and GR activities of germinating seeds of soybean exposed to a mixture of nano-SiO_2_ and nano-TiO_2_ remarkably promoted seed germination and seedling growth. However, the antioxidant enzyme activities of SOD, CAT, and GR in nanoparticle exposed plants vary significantly according to the plant species, nanomaterial type, duration of treatment, and dose.

## 5. Conclusions

ZnO NPs were synthesized successfully using leaf extracts of *O. arabicus.* The existence of phytochemicals in the leaf extract of *O. arabicus* helped in the synthesis of ZnO NPs by inducing oxidation and reduction reaction. The average size of the synthesized NPs was 35 nm. The positive effect of the synthesized ZnO NPs is apparent from the promoting of plant biomass. Our findings showed that application of green synthesized ZnO NPs in MO presented morpho-physiolgical changes where, the exposure of 5 mg L^−1^ ZnO NPs to the culture media significantly promoted shoot formation and enhanced the plant weight, level of photosynthetic pigments, and total protein content. The plant growth retreated with higher ZnO NPs doses (10 and 20 mg L^−1^), but increased the proline content and activated the production of antioxidant enzymes. On the other hand, MDA production was decreased in the plants.

This study indicates that the application of ZnO NPs to the in vitro culture media of plant tissues had positive influences; therefore, green-synthesized ZnO NPs hold promise for their judicious application in agriculture and medicine purposes. However, further investigation is indispensable for a better understanding of the molecular mechanism of ZnO NPs in cell developmental processes and secondary metabolism.

## Figures and Tables

**Figure 1 plants-10-01808-f001:**
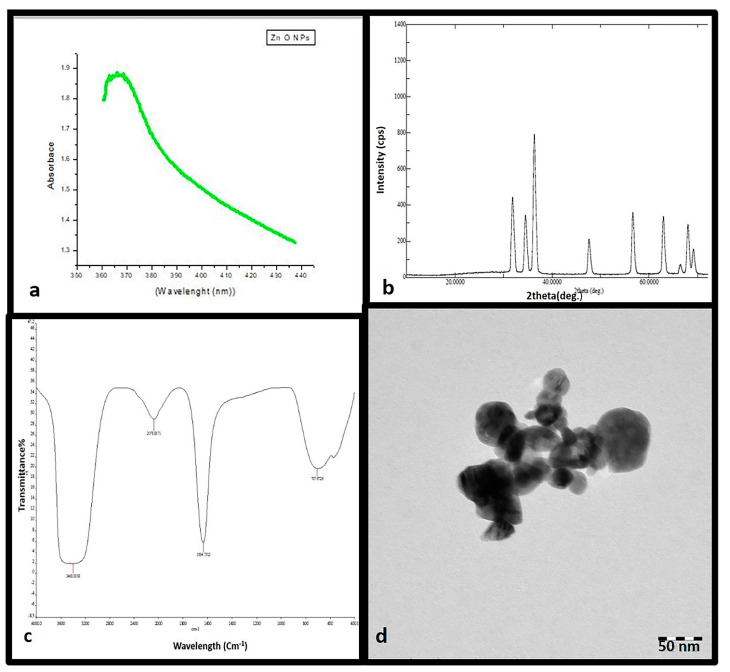
Ultraviolet–visible absorption spectrum of zinc oxide nanoparticles (ZnO NPs) with a plasmon band at 380 nm (**a**). XRD patterns (**b**). Fourier-transform infrared spectroscopy profile showing four peaks at 704, 1634, 2078, and 3400 cm^−1^(**c**). Transmission electron microscope image of ZnO NPs (**d**).

**Figure 2 plants-10-01808-f002:**
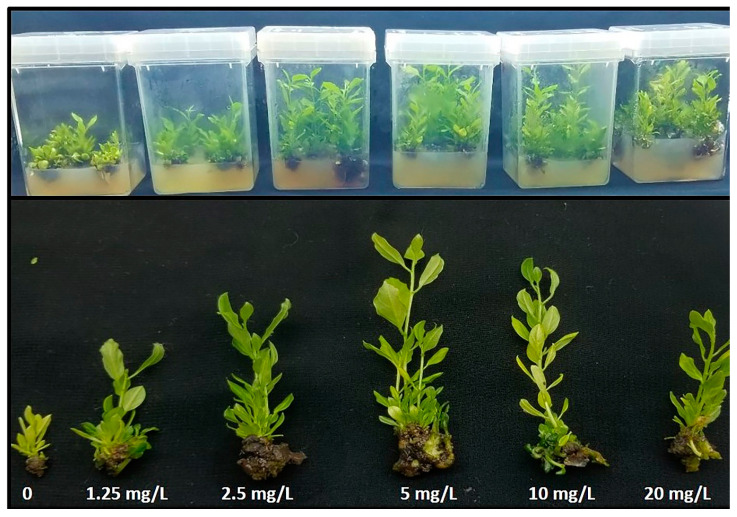
Effect of different concentrations of zinc oxide nanoparticles on in vitro multiplication of *Maerua oblongifoli*.

**Figure 3 plants-10-01808-f003:**
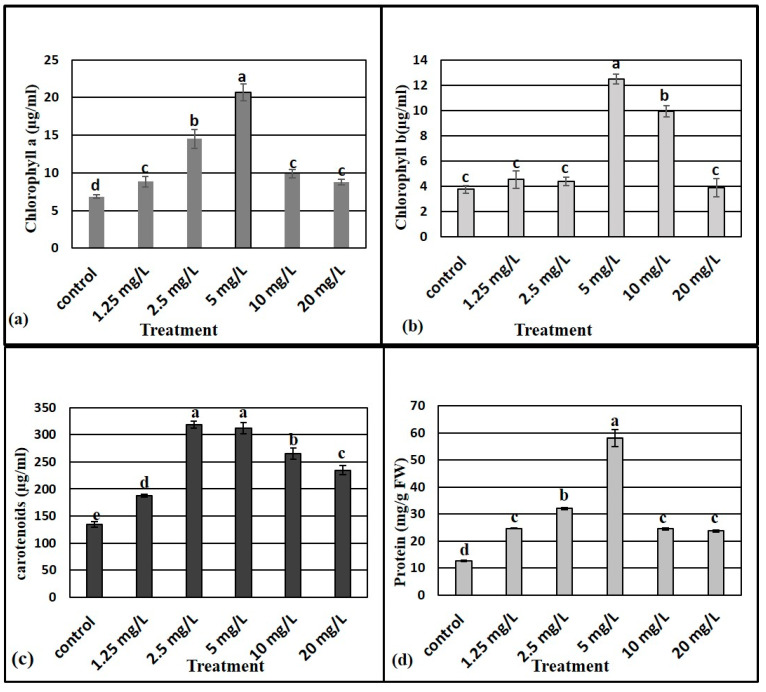
Effect of different concentrations of zinc oxide nanoparticles on the chlorophyll a (**a**), chlorophyll b (**b**), carotenoids (**c**) and protein (**d**) content of *Maerua oblongifolia*. Means ± SD for each concentration followed by the same letters are not significantly different according to one-way analysis of variance (*p* ≤ 0.05).

**Figure 4 plants-10-01808-f004:**
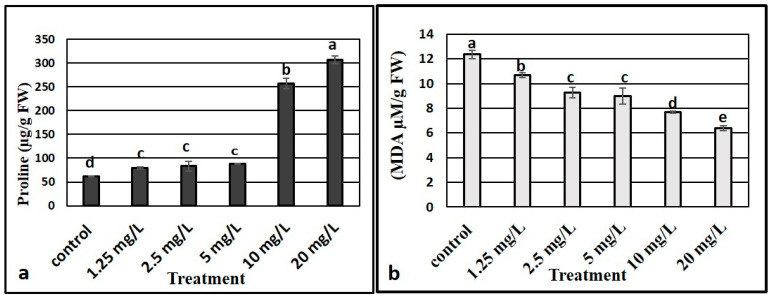
Effect of different concentrations of zinc oxide nanoparticles on proline (**a**) and MDA (**b**) content in *Maerua oblongifolia.* Means ± SD for each concentration followed by the same letters are not significantly different according to one-way analysis of variance (*p* ≤ 0.05).

**Figure 5 plants-10-01808-f005:**
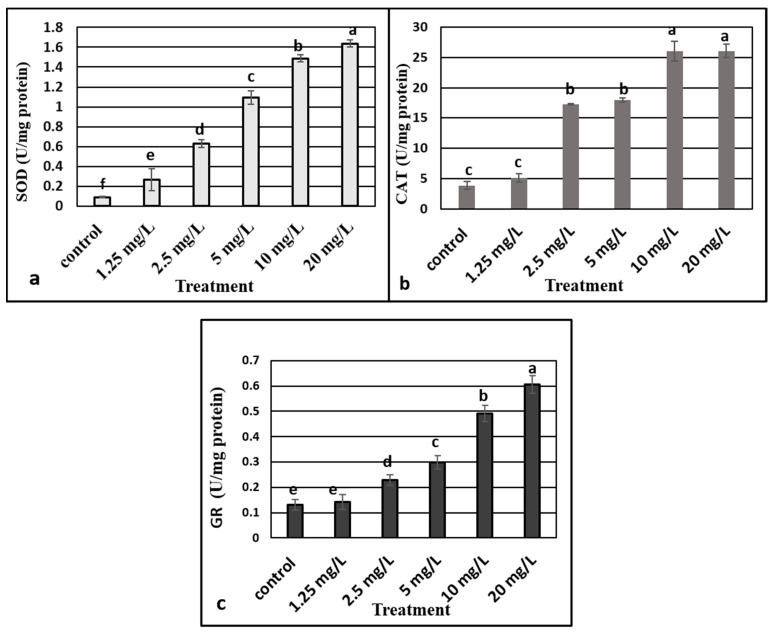
Effect of different concentrations of zinc oxide nanoparticles on superoxide dismutase (SOD) (**a**), catalase (**b**), and GR (**c**) activities in *Maerua oblongifolia.* Means ± SD for each concentration followed by the same letters are not significantly different according to one-way analysis of variance (*p* ≤ 0.05).

**Table 1 plants-10-01808-t001:** Effect of zinc oxide nanoparticles on the in vitro regeneration of *Maerua oblongifolia* after 45 days of treatment in MS media.

Treatment	Fresh Weight (g)	Dry Weight (g)	Shoot Number (Pot)	Shoot Length (cm)	Leaf Number (Pot)
control	5.03 ± 0.15 ^d^	0.73 ± 0.06 ^e^	11.67±0.58 ^e^	3.47 ± 0.06 ^f^	131.67 ± 0.58 ^d^
1.25 mg/L	7.45 ± 0.15 ^c^	1.73 ± 0.06 ^d^	16.67 ± 0.58 ^d^	5.07 ± 0.06 ^e^	162.67 ± 1.15 ^c^
2.5 mg/L	12.43 ± 0.15 ^b^	2.33 ± 0.06 ^c^	23.33 ± 0.58 ^b^	8.70 ± 0.10 ^c^	204.00 ± 1.15 ^b^
5 mg/L	13.2 ± 0.20 ^a^	2.90 ± 0.10 ^a^	26.67 ± 0.58 ^a^	11.23 ± 0.06 ^a^	213.67 ± 1.15 ^a^
10 mg/L	12.37 ± 0.06 ^b^	2.47 ± 0.06 ^b^	23.33 ± 0.58 ^b^	9.23 ± 0.06 ^b^	201.67 ± 1.15 ^b^
20 mg/L	7.53 ± 0.06 ^c^	1.67 ± 0.06 ^d^	21.00 ± 0.00 ^c^	6.17 ± 0.06 ^d^	171.33 ± 1.15 ^c^

The data representing the mean values of triplicates with ± standard deviation within a column followed by the same letters are not significantly different according to one-way analysis of variance (*p* ≤ 0.05).

## Data Availability

All data are presented within the article.

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
