# Peer review of "Biosynthesis and Characterization of ZnO Nanoparticles Using Ochradenus arabicus and Their Effect on Growth and Antioxidant Systems of Maerua oblongifolia"

_plants, 2021, doi:10.3390/plants10091808_

Round 1

Reviewer 1 Report

The article "Biosynthesis and characterization of ZnO nanoparticles using Ochradenus arabicus and their effect on growth and antioxidant system of Maerua oblongifolia" is written at the high scientific level in the top of the novelty. English is also understandable without any problems. The authors used a number of modern methods to describe the quality of the data provided and describe the characteristics of the nanoparticles.

But, there are some points that should be improved:

  • Fig. 1 a, b,c - please make a good quality of figs and add appropriate axis signature, where lost. For example, in Fig. 1c - there are no axis signatures at all.
  • I did not understand - what is the characterisation of ZnO NPS by IR - "Fourier transform infrared spectroscopy profile showing five peaks at 704, 1634, 2078, and 3400 cm−1"? What is the wavelength of ZnO emission? all above-listed wavelengths mostly characterized organic compounds or bounds. The same question to 1 b - please show the exact XRD spectra characterized your ZnO NPS. As it is shown in Fig 1b - it is not purely material. It has some additives and you cannot say that made a pure ZnO NPS treatment.
  • Fig. 1 d - the quality of the figure is appropriate, but the size in the microns looks like bad quality.
  • Figures 3, 4, 5 - please check the accuracy of the axis signatures. somewhere it is cut, somewhere there are mistakes. Treatmen - what does it mean? maybe additive or treatment?
  • Fig 1 d - Transmission 183
    electron microscope image of AgNPs (d). Is your article about Ag or Zn NPS?
  • It would be better to add some information in results, and widen discussion. Please be sure all discussed data form figures have significant difference.

Reviewer 2 Report

Dear Authors, in your interesting manuscript, the following points should be added/changed to further improve:

  1. Abstract: Please add information about the average size of the obtained ZnO nanoparticles.
  2. Introductions: I ask the authors to introduce an explanation of the abbreviation "NPs" used (34).
  3. Introductions: The “synthesis of ZnO” is one of the results reported in this work, therefore I suggest the authors add a few sentences of description regarding examples of zinc oxide nanoparticles synthesis and pointing out relevant references (preferably review papers).
  4. Introductions: I suggest the authors in describing the state of the art review also mention the works (DOI:10.3390/ma13122784, DOI:10.1016/j.eti.2021.101653, DOI:10.3390/agronomy11040729)
  5. Material and methods - Synthesis of ZnO NPs: Whether the reported method for ZnO synthesis is an author's method (own method)? Whether it was based on other work?
  6. Material and methods - Synthesis of ZnO NPs: Please correct the chemical formula of zinc acetate (67).
  7. Material and methods - Synthesis of ZnO NPs: Please indicate the volume of leaf extract used (69).
  8. Material and methods - Synthesis of ZnO NPs: I disagree with the following sentence “The color of the reaction mixture was changed to faint yellowish indicating the synthesis of ZnO NPs.” Do the authors have results confirming that they already obtained ZnO after the color change of the suspension at room temperature? After all, the color change may be a simple physical phenomenon resulting from the mixing of two suspensions of different color.
  9. Material and methods - Synthesis of ZnO NPs: Basic information (model, manufacturer) about the stirrer, oven and furnce is missing.
  10. Material and methods - Synthesis of ZnO NPs: Whether calcination of the sample took place in air?
  11. Material and methods - Characterization of ZnO NPs: Please provide basic information (model, manufacturer) on the analyzers used, such as Fourier transform infrared spectroscopy (FTIR), X-ray diffraction and transmission electron microscopy (TEM).
  12. Material and methods: Please explain the following abbreviations “MDA, NADPH” in the text of the manuscript.
  13. Results - Green synthesis and characterization of ZnO NPs: Once again, I disagree with the following conclusion “ZnO reduction with O. arabicus leaves extract caused a visible color change on stirrer for 24 h; the color change of the solution from white to pale yellow indicates the synthesis of ZnO NPs.” If the authors obtained ZnO at the very color change stage then why did they calcine the sample at 600C for 3 hours?
  14. Results - Green synthesis and characterization of ZnO NPs: Please explain to me what the authors meant by using the words “ZnO reduction (157)”.
  15. Results - UV-Vis spectroscopy: Please explain to me what the abbreviation “SPR” stands for (162) ?
  16. Results - XRD analysis: Why are the results of crystallites sizes not included in this section?
  17. Results - FTIR spectroscopy: Please explain to me what the authors intended to convey to readers in the sentence “The shift in the peaks was clearly confirmed the formation of ZnO nanoparticles. (174)” What shift is being referred to here?
  18. Results - Electron microscopy: I do not agree with the sentence “The TEM image revealed that the synthesized ZnO NPs were hexagonal in shape and size less than 50 nm, as given in Fig. 1d. 177-178”. After all, the TEM image (Fig. 1d) shows particles larger than 100 nm. Please explain this to me exactly why the authors claim to have obtained particles below 50 nm.
  19. Results - The name of Figure 1 is controversial “Figure 1. Ultraviolet-visible absorption spectrum of zinc oxide nanoparticles (AgNPs) with a plasmon band at 380 nm (a). XRD 182 patterns (b). Fourier transform infrared spectroscopy profile showing five peaks at 704, 1634, 2078, and 3400 cm−1. Transmission electron microscope image of AgNPs (d).” Please explain to me which AgNP the authors are referring to.
  20. Results - Effects of ZnONPs on in vitro shoot reproduction: Please correct the positions of the value descriptions of concentrations of ZnO NPs in Figure 2.
  21. Discussion: Please move the description of the use of the Scherrer equation to "Characterization of ZnO NPs".
  22. Discussion: Suggests rounding the value of the average crystallite size to an integer (252).
  23. Discussion: Please provide a few sentences of discussion regarding the comparison of particle size from the TEM method with average crystallite size from XRD method. Do the ZnO particles obtained consist of a single crystallite or several?
  24. Conclusion: Please add information about the average size of the obtained ZnO nanoparticles.

Round 2

Reviewer 2 Report

The answers from authors and the revised manuscript is acceptable at present form.

Author Response

No new Suggestions for Authors